

# Preoperative inflammatory markers of NLR and PLR as indicators of poor prognosis in resectable HCC

Dong Wang[1], Ning Bai[2], Xi Hu[1], Xi Wu OuYang[1], Lei Yao[1], YiMing Tao[1] and ZhiMing Wang[1]

[1] Department of General Surgery, Xiangya Hospital, Central South University, Hunan, China
[2] Department of Emergency, Xiangya Hospital, Central South University, Hunan, China

## ABSTRACT

**Background.** Many recent studies have demonstrated the predominant role chronic inflammation plays in cancer cell propagation, angiogenesis and immunosuppression. Cancer-related inflammation (CRI) has been shown to correlate with poor cancer prognosis. Our study aimed to evaluate the prognostic value of the neutrophil-to-lymphocyte ratio (NLR) and platelet-to-lymphocyte ratio (PLR) in patients with hepatocellular carcinoma (HCC) who have undergone liver resection.

**Methods.** Between 2012 and 2015, 239 patients with HCC who had undergone liver resection at XiangYa Hospital Central South University were included in this study. The values of simple inflammatory markers, including the NLR and PLR, used in predicting the long-term outcomes of these patients were evaluated using Kaplan–Meier curves and Cox regression models.

**Results.** The cutoff values of the NLR and PLR were 2.92 and 128.1, respectively. In multivariate Cox regression analysis, high NLR ($\geq$2.92) and high PLR ($\geq$128.1) were independent risk factors predicting poorer outcomes in patients with HCC. However, high NLR and high PLR were prognostic factors in tumor size and tumor number.

**Conclusions.** In this study, we identified that high NLR ($\geq$2.92) and high PLR ($\geq$128.1) are useful prognostic factors in predicting outcomes in patients with HCC whom underwent liver resection.

## INTRODUCTION

Hepatocellular carcinoma (HCC) is the most common type of cancer and the third leading cause of cancer-related death worldwide (*Ferlay et al., 2015*). Hepatitis infection plays a leading role in HCC occurrence and progression (*Bruix, Reig & Sherman, 2016*). Owing to a high occurrence of hepatitis B virus (HBV) and aflatoxin infection, China alone accounts for approximately half of all HCC cases, making HCC a major medical burden in our country. Hepatectomy and liver transplantation are considered as curative treatments for HCC patients (*Roayaie et al., 2015*; *Zhou et al., 2010*) and despite improved diagnosis and advances in surgical techniques, the clinical prognosis of HCC is still poor (*Villanueva et al., 2011*).

Corresponding authors
YiMing Tao, yimingtao@csu.edu.cn
ZhiMing Wang, zhimingwang@csu.edu.cn

Recently, many studies have demonstrated that chronic inflammation plays a predominant role in cancer cell propagation, angiogenesis and immunosuppression (*Chaturvedi et al., 2010*). Cancer-related inflammation (CRI) has been shown to correlate with poor cancer prognosis (*Elinav et al., 2013*; *Antonioli et al., 2013*). Inflammation caused by EB virus infection is related to nasopharyngeal cancer, hepatitis virus infection leads to HCC, and Helicobacter pylori infection leads to gastric cancer. CRI helps cancer cells to acquire malignant biological behaviors, including proliferation, infiltration, angiogenesis, and metastasis. The nuclear factorκb (NF-κB) (*Ratnam et al., 2017*) and transcription activator 3 (STAT3) (*Izumi et al., 2013*) pathways are well known in CRI. Chemokines, including TNF (*Balkwill, 2009*), CXCL8 (*Manfroi et al., 2017*), and IL-6 (*He et al., 2013*), also play an important role in the pathophysiological process of tumor formation. CRI parameters, including C-reactive protein (CRP) (*Chaturvedi et al., 2010*), platelet-to-lymphocyte ratio (PLR) (*Dalpiaz et al., 2017*), and neutrophil-lymphocyte ratio (NLR) (*McNamara et al., 2014*), are widely used in cancer patients to guide treatment and predict prognosis. These biomarkers are more readily available and non-invasive.

However, the ability of the NLR and PLR to predict the prognosis of patients with HCC after liver resection is under debate. Our study was designed to combine the preoperative inflammatory markers NLR and PLR to evaluate the prognosis of patients with HCC whom underwent curative resection.

## MATERIALS & METHODS

### Study population

Our study included 239 patients with HCC whom underwent liver resection between 2012 and 2015 at XiangYa Hospital, Central South University, China. HCC was confirmed using postoperative pathology. Patients with any one of the following items were excluded from this study: (1) had undergone splenectomy; (2) recurrence of HCC; (3) ruptured HCC; (4) infections during the perioperative period; (5) other autoimmune diseases; (6) preoperative antitumor treatments; and (7) preoperative application of interferon, interleukin or other similar drugs. This study was approved by the ethics committee of XiangYa Hospital Central South University (No. 201709984) and was conducted with the patients' informed consent.

### Follow-up and definitions

Blood routine, liver function, serum alpha-fetoprotein (AFP), and hepatitis B surface antigen (HBsAg) were tested in all patients. Abdomen ultrasonography, computed tomography (CT) or magnetic resonance imaging (MRI), and chest radiography were performed for all patients. NLR measured neutrophil count to lymphocyte count, and PLR measured platelet count to lymphocyte count. Recurrence was diagnosed using imaging (CT or MRI) and AFP. An AFP level >20 ng/mL was defined as being high (*Tao et al., 2013*). The cutoff values of NLR and PLR were determined using receiver operating characteristic curves (ROC) according to the overall survival of patients. The seventh edition of the American Joint Committee on Cancer tumor-node metastasis (TNM) staging system and Barcelona Clinic Liver Cancer were applied to rank the HCC stage.

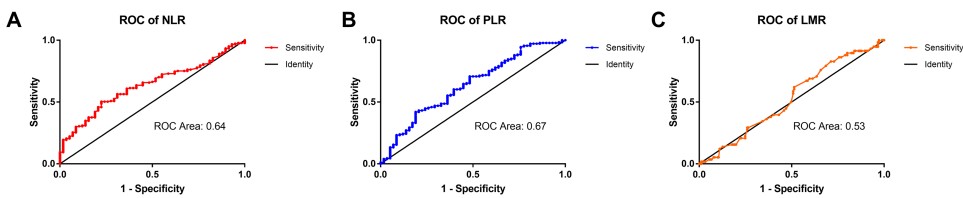

**Figure 1  The ROC curves of the NLR, PLR, and LMR in patients with HCC.** (A) The ROC area of NLR was 0.63. (B) The ROC area of PLR was 0.67. (C) The ROC area of LMR was 0.53.

## Statistical analysis

Statistical analyses were performed using Prism software (GraphPad Prism Software, La Jolla, CA, USA) and SPSS 21.0 (SPSS Company, Chicago, IL, USA) for Windows. Quantitative values were analyzed using $t$ tests. Categorical variables were compared using the chi-square test or Fisher's exact test. Recurrence-free survival (RFS) and overall survival (OS) were evaluated using the Kaplan–Meier method and the log-rank test. Prognostic factors of RFS and OS were analyzed using univariate and multivariate analyses (*Hu et al., 2018*). $P < 0.05$ was considered statistically significant.

## RESULTS

### Assessment of the cut-off value of NLR, PLR and LMR

According to the ROC curve, the ideal cutoff values for preoperative NLR and PLR were 2.92 and 128.1, respectively. The ROC areas under the curve for the NLR and PLR were 0.63 (95% CI for the area between 0.56 to 0.71) and 0.67 (95% CI for the area between 0.55 to 0.72), respectively. The cutoff values of the NLR and PLR presented correspond to sensitivity values of 51% and 81%, and specificity values of 78% and 42%, respectively (Fig. 1).

### The relationship of clinical and pathologic characteristics with preoperative NLR and PLR in patients with HCC

A total of 239 patients met the enrollment conditions, including 200 (83.68%) males and 39 (16.32%) females, and were enrolled in the present study. As presented in Table 1, the mean age was (50.14 ± 11.98) years. The mean tumor size was (5.88 ± 3.59) cm and 57 (23.85%) patients had multiple tumors. A high preoperative AFP was observed in 155 (64.85%) patients. HBV surface antigen was positive in 202 (84.5%) patients, 71 (29.7%) patients had tumor encapsulation, and 174 (72.8%) patients had liver cirrhosis that was confirmed by pathology.

As shown in Table 1, the relationships between preoperative NLR and PLR, and between clinical and pathologic characteristics were investigated. The high-NLR group included 104 (43.51%) patients (NLR > 2.92) and 135 (56.49%) patients identified as being in the low-NLR (NLR ≤ 2.92) group. Eighty-seven (36.4%) patients were identified as being in the high-PLR group (PLR > 128.1), and 152 (63.6%) patients were identified as being in the low-PLR group (PLR ≤ 128.1).

**Table 1  HCC patients ($n = 239$) categorized by NLR, PLR and their clinical pathologic characteristics.**

| Clinical character | | NLR | | | PLR | | |
|---|---|---|---|---|---|---|---|
| | | ≤2.92 ($n = 135$) | >2.92($n = 104$) | *P*-value | ≤128.1($n = 152$) | >128($n = 87$) | *P*-value |
| Age, years | | 49.0 ± 12.47 | 51.28 ± 11.33 | 0.34 | 48.82 ± 11.21 | 52.54 ± 13.01 | 0.02 |
| Serum albumin, g/L | | 41.36 ± 0.38 | 41.49 ± 0.5 | 0.59 | 41.82 ± 0.39 | 40.97 ± 0.49 | 0.18 |
| Tumor size, cm | | 5.01 ± 0.26 | 6.99 ± 0.36 | 0.00 | 5.18 ± 0.25 | 7.17 ± 0.39 | 0.00 |
| Platelet 10^9/L | | 156.0 ± 6.92 | 167.7 ± 6.89 | 0.25 | 133.4 ± 4.43 | 209.7 ± 9.03 | 0.00 |
| TBil, μmol/L | | 14.2 ± 0.63 | 16.24 ± 1.52 | 0.18 | 14.56 ± 0.57 | 16.02 ± 1.82 | 0.35 |
| ALT, U/L | | 41.06 ± 2.77 | 44.91 ± 3.19 | 0.36 | 41.66 ± 2.39 | 44.62 ± 3.95 | 0.49 |
| AST, U/L | | 44.51 ± 2.72 | 50.18 ± 3.05 | 0.17 | 44.18 ± 2.14 | 52.28 ± 4.12 | 0.06 |
| PT, s | | 13.22 ± 0.10 | 13.33 ± 0.10 | 0.57 | 13.38 ± 0.09 | 13.06 ± 0.09 | 0.02 |
| Gender | Male | 115 | 85 | 0.49 | 132 | 68 | 0.10 |
| | Female | 20 | 19 | | 20 | 19 | |
| HBsAg | Negative | 23 | 14 | 0.48 | 19 | 18 | 0.10 |
| | Positive | 112 | 90 | | 133 | 69 | |
| AFP, ng/mL | ≤20 | 50 | 34 | 0.49 | 45 | 39 | 0.02 |
| | >20 | 85 | 70 | | 107 | 48 | |
| Liver cirrhosis | No | 33 | 32 | 0.31 | 41 | 24 | 0.92 |
| | Yes | 102 | 72 | | 111 | 63 | |
| Tumor encapsulation | No | 96 | 72 | 0.78 | 110 | 58 | 0.37 |
| | Yes | 39 | 32 | | 42 | 29 | |
| Tumor number | Single | 105 | 77 | 0.54 | 119 | 63 | 0.35 |
| | Multiple | 30 | 27 | | 33 | 24 | |
| Satellite nodules | No | 125 | 96 | 0.98 | 146 | 75 | 0.01 |
| | Yes | 10 | 8 | | 6 | 12 | |
| Edmondson grade | I–II | 103 | 83 | 0.53 | 116 | 70 | 0.52 |
| | III–IV | 32 | 21 | | 36 | 17 | |
| BCLC stage | 0 | 11 | 0 | 0.00 | 10 | 1 | 0.04 |
| | A | 93 | 60 | | 101 | 52 | |
| | B | 28 | 17 | | 27 | 18 | |
| | C | 3 | 27 | | 14 | 16 | |
| TNM stage | I | 95 | 48 | 0.00 | 102 | 41 | 0.01 |
| | II | 30 | 19 | | 27 | 22 | |
| | III | 10 | 37 | | 23 | 24 | |

**Notes.**

**Characteristics.**

NLR, neutrophil-to-lymphocyte ratio; HBsAg, hepatitis B surface antigen; AFP, α-fetoprotein; TNM, tumor-node-metastasis; TBil, total bilirubin; PT, prothrombin time; CTP, Child-Turcotte-Pugh; BCLC stage, The Barcelona Clinic Liver Cancer staging; ALT, glutamic-pyruvic transaminase; AST, glutamic oxalacetic transaminase.

Preoperative NLR level and PLR level were closely correlated with tumor size, TNM stage and BCLC stage ($P < 0.05$). The PLR also correlated with age, platelet count, prothrombin time (PT), AFP, and satellite nodules ($P < 0.05$). No obvious correlations with gender, HBsAg, liver cirrhosis, serum albumin, total bilirubin (TBil), glutamic-pyruvic transaminase (ALT), or glutamic-oxaloacetic transaminase (AST) were observed ($P > 0.05$).

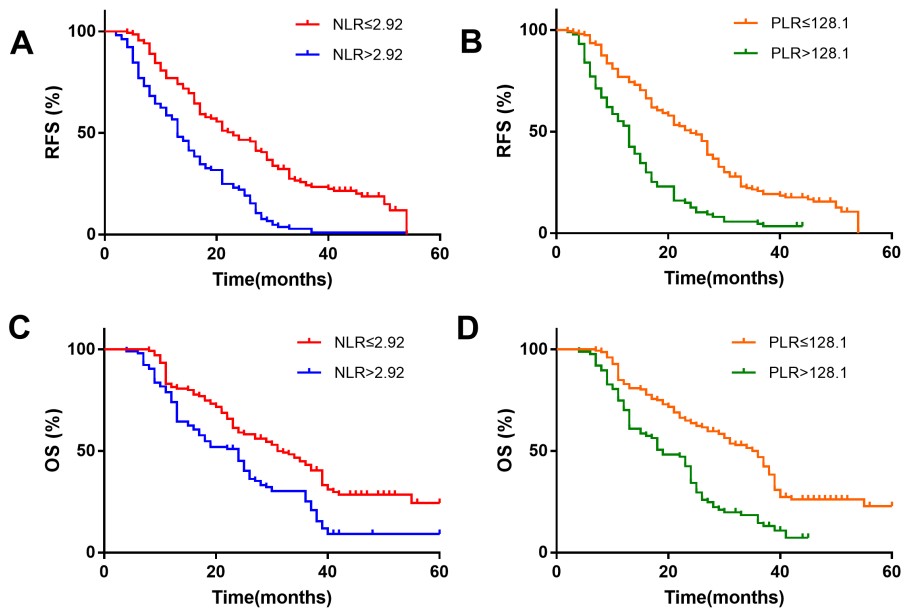

**Figure 2** Kaplan–Meier survival analysis indicates that patients with NLR > 2.92 have a shorter RFS and OS (A and C); patients with PLR > 128.1 have a shorter RFS and OS (B and D).

## The correlation between NLR, PLR and postoperative RFS and OS in patients with HCC who underwent liver resection

Kaplan–Meier survival analysis showed that the NLR>2.92 group was associated with a shorter recurrence-free survival (RFS) (Fig. 2A) and overall survival (OS) (Fig. 2C). The patients with HCC in the PLR>128.1 group were also associated with a shorter RFS (Fig. 2B) and OS (Fig. 2D).

From the univariate analysis in Table 2, we found that tumor size (HR 1.30, 95% CI [1.08–1.56]), NLR (HR 2.85, 95% CI [1.63–4.93]), PLR (HR 1.013, 95% CI [1.00–1.02]), BCLC stage (HR 3.005, 95% CI [1.39–6.50]), and satellite nodules (HR 4.27, 95% CI [2.55–7.14]) correlated with RFS in patients with HCC who underwent liver resection ($P<0.05$). The platelet count (HR 1.01, 95% CI [1.00–1.01]), AST (HR 1.02, 95% CI [1.00–1.03]), tumor size (HR 1.42, 95% CI [1.23–1.63]), NLR (HR 1.48, 95% CI [1.16–1.88]), PLR (HR 1.007, 95% CI [1.001–1.013]), TNM stage (HR 19.42, 95% CI [2.61–144.3]), BCLC stage (HR 2.43, 95% CI [0.98–5.98]), satellite nodules (HR 4.42, 95% CI [2.66–7.33]), and tumor number (HR 2.78, 95% CI [1.18–6.54]) correlated with OS ($P < 0.05$). Gender, HBsAg, liver cirrhosis, serum albumin, total bilirubin (TBil), ALT and so on had no statistically significant association with RFS or OS ($P > 0.05$).

In the multivariate analysis, we found that NLR (HR 1.16, 95% CI [1.06–1.26]) and PLR (HR 1.01, 95% CI [1.001–1.006]) were independent risk factors for RFS in patients with HCC. NLR (HR 1.14, 95% CI [1.04–1.25]) and PLR (HR 1.004, 95% CI [1.001–1.007]) were independent risk factors for OS in patients with HCC.
**Table 2** Univariate and multivariate analyses of prognostic factors with RFS and OS in patients with HCC (*n* = 239).

| Clinicopathologic variable | RFS HR (95% CI) | *P*-value | OS HR (95% CI) | *P*-value |
|---|---|---|---|---|
| **Univariate analysis** | | | | |
| Gender (male vs. female) | 2.40 (0.54–10.64) | 0.25 | 1.93 (0.77–4.89) | 0.16 |
| Age, years (>60 vs. ≤60) | 1.00 (0.97–1.04) | 0.86 | 0.98 (0.96–1.00) | 0.21 |
| Serum albumin, g/L (≤35 vs. >35) | 1.00 (0.88–1.04) | 0.28 | 0.99 (0.93–1.05) | 0.73 |
| Platelet,10^9/L (≤160 vs. >160) | 1.00 (0.99–1.008) | 0.50 | 1.01 (1.001–1.01) | 0.02 |
| TBil, μmol/L (≤17.1 vs. >17.1) | 1.02 (0.97–1.05) | 0.83 | 1.01 (0.98–1.05) | 0.52 |
| ALT, U/L (≤50 vs. >50) | 1.01 (0.99–103) | 0.40 | 1.01 (0.98–1.03) | 0.09 |
| AST, U/L (≤40 vs. >40) | 1.01 (0.99–1.03) | 0.29 | 1.02 (1.00–1.03) | 0.05 |
| PT, s (≤13.2 vs. >13.2) | 0.89 (0.60–1.31) | 0.54 | 0.82 (0.61–1.08) | 0.16 |
| AFP, ng/mL (>20 vs. ≤20) | 2.18 (0.95–5.03) | 0.07 | 1.72 (0.94–3.15) | 0.08 |
| HBV (presence vs. absence) | 4.86(0.64–37.04) | 0.13 | 1.19(0.54–2.63) | 0.67 |
| NLR (>2.92 vs. ≤2.92) | 2.85 (1.63–4.93) | <0.01 | 1.48 (1.16–1.88) | <0.01 |
| PLR (>128.1 vs. ≤128.1) | 1.01 (1.00–1.02) | 0.012 | 1.01 (1.00–1.013) | 0.014 |
| BCLC stage (C vs. 0/A/B) | 3.01 (1.39–6.50) | <0.01 | 2.43 (0.98–5.98) | <0.01 |
| TNM stage (II/III vs. I) | 6.57 (0.87–49.8) | 0.01 | 19.42 (2.61–144.3) | <0.01 |
| Tumor number (multiple vs. single) | 2.48 (0.71–8.56) | 0.15 | 2.78 (1.18–6.54) | 0.02 |
| Edmondson grade (III/IV vs. I/II) | 1.56 (0.51–4.76) | 0.44 | 1.12 (0.54–2.32) | 0.75 |
| Tumor size, cm (>5 vs. ≤5) | 1.30 (1.08–1.57) | <0.01 | 1.42 (1.23–1.63) | <0.01 |
| Satellite nodules (presence vs. absence) | 4.27 (2.55–7.14) | <0.01 | 4.42 (2.66–7.33) | <0.01 |
| Tumor encapsulation (none vs. complete) | 1.34 (0.53–3.62) | 0.51 | 1.45 (0.73–2.85) | 0.29 |
| Liver cirrhosis (presence vs. absence) | 1.30 (0.53–3.17) | 0.57 | 1.03 (0.53–1.99) | 0.94 |
| Hospital stay, d | 1.02 (0.88–1.19) | 0.75 | 1.04 (0.94–1.16) | 0.45 |
| **Multivariate analysis** | | | | |
| Platelet, 10^9/L (≤160 vs. >160) | NA | | 0.99 (0.99–1.00) | 0.46 |
| AST, U/L (≤40 vs. >40) | NA | | 1.00 (0.99–1.01) | 0.42 |
| AFP, ng/mL (>20 vs. ≤20) | 1.39 (1.05–1.88) | 0.03 | 1.37 (1.01–1.86) | 0.04 |
| Tumor size, cm (>5 vs. ≤5) | 1.10 (1.05–1.15) | 0.01 | 1.10 (1.05–1.16) | 0.01 |
| NLR (>2.92 vs. ≤2.92) | 1.16 (1.06–1.26) | <0.01 | 1.14 (1.04–1.25) | <0.01 |
| PLR (>128.1 vs. ≤128.1) | 1.01 (1.00–1.01) | <0.01 | 1.004 (1.00–1.01) | <0.01 |
| TNM (II/III vs. I) | 1.40 (0.77–2.53) | 0.27 | 1.39 (0.76–2.55) | 0.28 |
| Tumor number (multiple vs. single) | 1.34 (0.92–1.96) | 0.13 | 1.33 (0.91–1.95) | 0.15 |
| Satellite nodules (presence vs. absence) | 3.03 (1.62–5.65) | 0.00 | 2.98 (1.59–5.57) | 0.00 |

**Notes.**

NLR, neutrophil-to-lymphocyte ratio; HBsAg, hepatitis B surface antigen; AFP, α-fetoprotein; TNM, tumor-node-metastasis; TBil, total bilirubin; PT, Prothrombin time; CTP, Child-Turcotte-Pugh; BCLC stage, The Barcelona Clinic Liver Cancer staging; ALT, glutamic-pyruvic transaminase; AST, glutamic oxalacetic transaminase.

## Combined NLR and PLR to analyze RFS and OS in patients with HCC who underwent hepatectomy

In the previous results, we found that high NLR and high PLR are independent risk factors for RFS and OS after hepatectomy in patients with HCC. We combined NLR with PLR to investigate whether the prediction of RFS and OS was more accurate. We defined NLR ≤2.92 as NLR$_{-low}$, NLR>2.92 as NLR$_{-high}$, PLR ≤128.1 as PLR$_{-low}$, and PLR>128.1 as PLR$_{-high}$. We found that patients with simultaneously high NLR and PLR had the worst

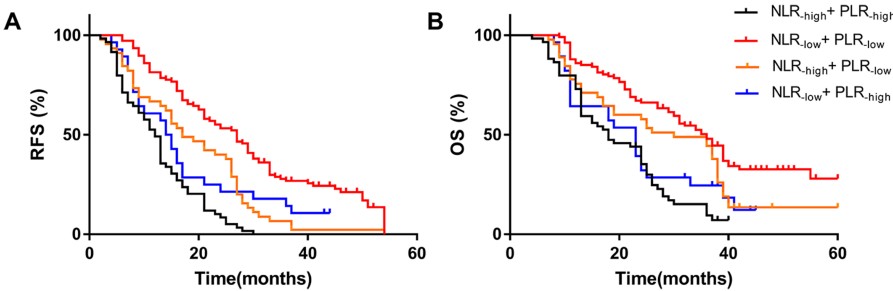

**Figure 3** **Effect of combined NLR and PLR on RFS and OS in HCC patients whom underwent hepatectomy.** The RFS of combined NLR and PLR(A) and the OS of combined NLR and PLR(B). The NLR-high and PLR-high group had the worst RFS (median 12 months) and OS (median 18 months), and the NLR-low and PLR-low group had the best RFS (median 14.5 months) and OS (median 23 months).

RFS (median 12 months) and OS (median 18 months), while patients with simultaneously low NLR and PLR had the best RFS (median 14.5 months) and OS (median 23 months). The $NLR_{-low}$ and $PLR_{-low}$ groups had the best outcome and their RFS and OS were superior to other groups. The worst group was the $NLR_{-high}$ group combined with the $PLR_{-high}$ group. The results showed that patients with simultaneously high NLR and high PLR were more prone to metastasis and had the worst OS (Fig. 3).

**The relationship between NLR, PLR, tumor size, and satellite nodules**
Using multivariate analysis, we found that tumor size was an independent risk factor for poor prognosis in patients with HCC who underwent liver resection. To see if there is any correlation, we analyzed the relationship between NLR, PLR and tumor size. We divided the tumors into three groups by size: ≤3 cm group, between 3–10 cm group, and ≥10 cm group. We found the NLR and PLR were higher in groups with larger tumor size ($P < 0.05$) (Fig. 4).

The mean NLRs in the tumor ≤3 cm group, 3–10 cm group and ≥10 cm group were ($2.32 \pm 0.15$), ($3.23 \pm 0.17$), and ($4.03 \pm 0.38$), respectively (Fig. 4A); the mean PLRs were ($90.21 \pm 6.44$), ($128.5 \pm 5.4$), ($157 \pm 13.41$), respectively (Fig. 4B). We hypothesized that with high neutrophil and platelet counts, cancer cells can release various chemokines and promote tumor growth. At the same time, the number of lymphocytes decreased, and tumor cells escaped from the immune surveillance, as the immune system could not activate its normal anti-tumor effect. As a result, the HCC tumor growth progresses, increasing the tumor size.

We further analyzed the relationship between NLR, PLR and BCLC stage. We found that the advantaged BCLC stage had higher NLR and PLR values. The mean NLR values of BCLC 0, A, B, and C stages were ($1.70 \pm 0.14$), ($2.93 \pm 0.13$), ($3.05 \pm 0.26$), and ($4.82 \pm 0.65$) (Fig. 5A), respectively. The mean PLR values of BCLC 0, A, B, and C stages were ($81.93 \pm 10.68$), ($122.1 \pm 5.61$), ($122.5 \pm 8.91$), and ($149.2 \pm 16.13$), respectively (Fig. 5B).

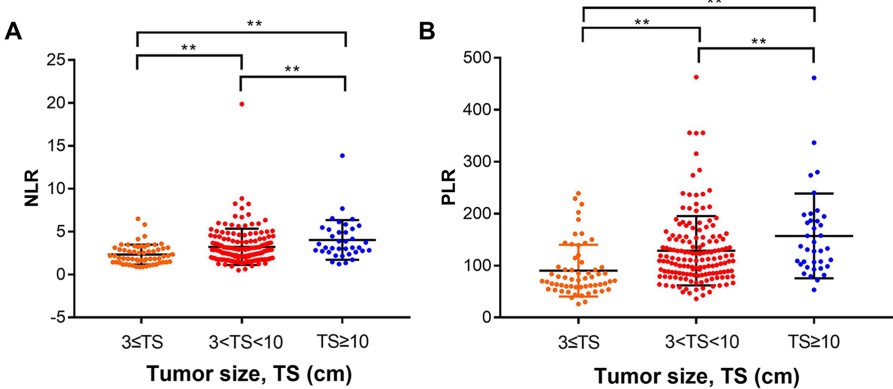

**Figure 4** **Analysis of the relationship between NLR, PLR and tumor size.** We divided the tumors into three groups according to size: ≤ three cm group, between 3–10 cm group, and ≥10 cm group. (A) The mean NLRs in the tumor ≤ three cm group, 3–10 cm group and ≥ 10 cm group were (2.32 ± 0.15), (3.23 ± 0.17), and (4.03 ± 0.38), respectively. (B) the mean PLRs were (90.21 ± 6.44), (128.5 ± 5.4), and (157 ± 13.41), respectively. ** means $P < 0.05$.

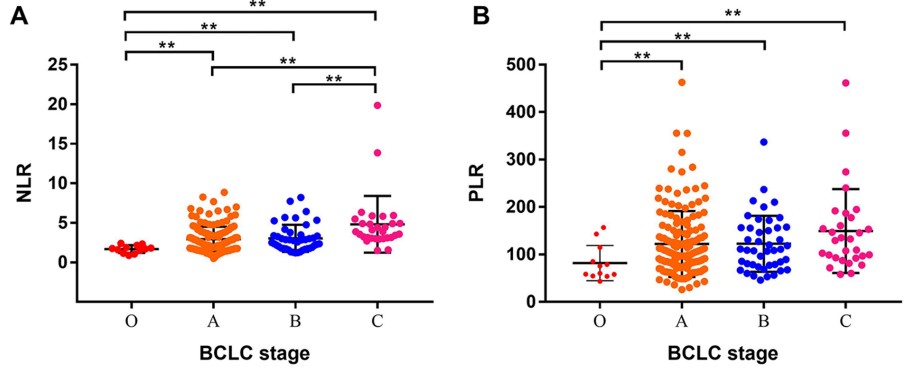

**Figure 5** **Analysis of the relationship between NLR, PLR and BCLC stage.** (A) The mean NLR values of the BCLC 0, A, B, and C stages were (1.70 ± 0.14), (2.93 ± 0.13), (3.05 ± 0.26), and (4.82 ±0.65), respectively. (B) The mean PLR values of the BCLC 0, A, B, and C stages were (81.93 ± 10.68), (122.1 ± 5.61), (122.5 ± 8.91), and (149.2 ± 16.13), respectively. ** means $P < 0.05$.

## DISCUSSION

Many researchers have demonstrated that inflammation contributes to the pathogenesis and progression of cancer (*Sanford et al., 2013*). The presence of systemic inflammation is associated with poor survival in many types of tumors, and anti-inflammatory agents have been associated with cancer prevention and treatment (*Pribluda et al., 2013*). Inflammation can promote cancer development through multiple mechanisms, including gene mutation, cancer cell proliferation and angiogenesis (*Elinav et al., 2013*). NLR and PLR have been shown to have the ability to predict the prognosis in various cancers, including HCC (*Lu et al., 2016*), esophageal carcinoma(*Feng, Huang & Chen, 2014*), renal carcinoma (*Hu et al., 2017*), and lung cancer (*Sanchez-Salcedo et al., 2016*). *Lu et al. (2016)* have studied the

NLR in early and intermediate stage HCC, and in our research, we found the NLR and PLR can predict the prognosis of patients with HCC who underwent liver research. The stages included were early, intermediate and advanced stage HCC, all showing similar results.

In solid tumors, inflammation often appears before the tissue malignant transformation. The occurrence and development of systemic immune responses provide an appropriate microenvironment for cancer metastasis and recurrence. In China, most patients with HCC have hepatitis infection, the inflammatory status playing an important role in promoting the development of HCC. The NLR and PLR, sensitive indexes of the body's inflammation system, can reflect the inflammatory state and predict the prognosis of the tumor.

Neutrophil can strengthen the biological behavior of the tumor, causing it to grow and metastasize. Higher neutrophil levels can upregulate the expression of growth factors, such as the types of chemokines, which play an important role in tumor development and progression. Platelets play a leading role in tumor progression. Platelets can secrete inflammatory factors, including TGF-β and VEGF, which can accelerate the differentiation and proliferation of tumor cells. Moreover, platelets release platelet derived factors, such as platelet reactive protein, etc., which play an important role in tumor adhesion, and angiogenesis to (1) prepare the microenvironment for tumor metastasis by secreting angiogenic factor and growth factor; and (2) shield the cancer cell so the platelets can adhere to the tumor. Platelets can protect cancer cells from the mechanical force of blood flow, and can also provide a shield for cancer cells that allows them to escape immune surveillance.

Many studies have confirmed that lymphocytes are the most important cells in tumor killing. When there is a relative or absolute reduction of lymphocytes, the antitumor effect is also decreased. PD-1 and CTLA–4 inhibitors are the most important immunity drugs (*Rizvi et al., 2015*; *Wei et al., 2017*). They can reduce tumor cell and T lymphocyte cell interaction by inhibiting the cancer cell surface expression of PD-1 and CTAL-4. The patient's decreased immunity, specifically the abnormality of the tumor immune microenvironment, leads to the failure of the lymphocyte immune response, and the cancer cells escape from immune surveillance. When immune tolerance or immune escape occurs, tumors are more likely to progress or metastasize. In our study, we found that patients with higher NLR and PLR had worse RFS and OS prognosis. On one hand, the increase of neutrophil and platelet counts promotes tumorigenesis; on the other hand, the decrease in the number of lymphocytes leads to the patient's immunity decline, leading to tumor progression.

In our study, we found that tumor size correlated with the NLR and PLR; the larger the tumor size, the higher the NLR and PLR. We hypothesize that (1) as neutrophil and platelet counts increase, they secrete many kinds of growth factors and inflammatory factors, which promote the growth of tumor cells and stromal cells and impact the tumor microenvironment and promote tumor growth; and (2) larger tumor size means higher tumor burden. The number of lymphocytes obviously decreased, and the effect of tumor cells killing is also weakened, thereby promoting the development of the tumor. Tumor size is one of the prognostic predictors for patients with HCC, but tumor size was more difficult to measure than the NLR and PLR. Additionally, tumor size cannot provide an accurate

prediction of HCC, because the NLR and PLR can reflect whether patients with HCC are associated with cirrhosis and hypersplenism. If the tumor is small, but liver cirrhosis and hypersplenism are obvious, the prognosis of patients with HCC will be poor.

We also found that the BCLC stage correlated with high NLR and high PLR. The advanced BCLC stages had higher PLR and NLR. Multiple tumors and/or vascular invasion in patients with HCC may lead to a stronger inflammatory response and weaker immune response. Higher neutrophil and platelet counts mean a stronger inflammatory response. Lower lymphocyte counts mean the immune response is decreased, and cancer cells are more likely to metastasize.

This study has some limitations. First, the number of patients in our study is small and the patients were retrospectively studied in a single center. Therefore, we could not avoid selection bias when collecting information on patients with HCC. Second, the NLR and PLR were assessed by single measurements at the time of admission for the initial diagnosis.

## CONCLUSION

In conclusion, our study showed that NLR and PLR are useful prognostic factors in predicting outcomes in patients with HCC who underwent live resection. This finding can assist in guiding the clinical management of patients with HCC.

## ACKNOWLEDGEMENTS

We gratefully acknowledge all of the authors' work on this paper and all of the patients in our study.

### Funding

This study was supported by grants from the National Nature Science Foundation of China (No. 81372631, 81372630). The funders had no role in study design, data collection and analysis, decision to publish, or preparation of the manuscript.

### Grant Disclosures

The following grant information was disclosed by the authors:
National Nature Science Foundation of China: 81372631, 81372630.

### Competing Interests

The authors declare there are no competing interests.

### Author Contributions

- Dong Wang performed the experiments, prepared figures and/or tables.
- Ning Bai and Xi Hu analyzed the data.
- Xi Wu OuYang analyzed the data, contributed reagents/materials/analysis tools.
- Lei Yao contributed reagents/materials/analysis tools.

- YiMing Tao conceived and designed the experiments, performed the experiments, authored or reviewed drafts of the paper.
- ZhiMing Wang conceived and designed the experiments, approved the final draft, write this paper.

## Human Ethics

The following information was supplied relating to ethical approvals (i.e., approving body and any reference numbers):

This study was approved by the ethics committee of Xiangya Hospital Central South University (No. 201709984).

## Data Availability

The raw data is available as a Supplemental File.

## Supplemental Information

Supplemental information for this article can be found online at http://dx.doi.org/10.7717/peerj.7132#supplemental-information.

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
