# Peer review of "Preoperative inflammatory markers of NLR and PLR as indicators of poor prognosis in resectable HCC"

_PeerJ, doi:10.7717/peerj.7132_

## Round 0.1 · original submission · Major Revisions

Please address all concerns of the reviewers mentioned in comments for the Author.

Reviewer 1 ·

Basic reporting

This manuscript provied the clearly inforamation of NLR and PLR in the HCC patients, the research background was shown in the paper and the literature well referenced. The authors have found that the NLR and PLR are useful prognostic factors to predict outcomes in patients with HCC who underwent live resection. The manuscript structure conforms to Peer J standards, and the authors uploaded the research raw data.

Experimental design

In China, the hepatitis infection plays a leading role in HCC occurrence and progression, the researchers aimed to found new biomarker in the HCC patients who underwent the liver resection, in order to assisted to guide treatment and predict prognosis. The paper’s question was started with HCC clinical and was well defined.
The experimental design was rigorous, the idea of the paper was concise and the experimental results are reliable.

Validity of the findings

Inflammation palys great leading role in the pathogenesis and progression of cancer. Neutrophil count promotes tumor growth and metastasis and the mechanism may included extracellular matrix, reactive oxygen species. Platelets also can assist the tumor progression. Platelets can secrete inflammatory factors, including TGF-β and VEGF.
In this paper, the authors found that the preoperative NLR level and PLR level were closely correlated with the tumor size, TNM stage and BCLC stage, high NLR and high PLR were had the worst outcomes. The findings in this paper would assist the surgeons development the individualized management for the HCC patients.

Additional comments

This is a well-written paper containing credible results which merit publication. For the benefit of the reader, however, a number of points need clarifying and certain statements require further justification. There are given below:
1. In the lines 45, the description “hepatitis virus infection leads to liver cancer” should be descripted preciseness.

2. In the study population section, the recurrence of HCC patients who underwent liver resection were excluded or not?

3. The English language should be improved to ensure that an international audience can clearly understand your text.

4. The following study should be cited and disscued.
Preoperative Ratio of Neutrophils to Lymphocytes Predicts Postresection Survival in Selected Patients With Early or Intermediate Stage Hepatocellular Carcinoma. Medicine (Baltimore). 2016;95(5):e2722. doi: 10.1097/MD.0000000000002722.

·

Basic reporting

no comment

Experimental design

no comment

Validity of the findings

no comment

Additional comments

The authors analyzed the relationship between preoperative NLR,PLR and RFS,OS from 239 HCC patients with liver resection and came to the result that NLR and PLR could be independent prognostically predictor.
This manuscript is not original since there are already similar studies analyzing the NLR and PLR predicting the prognosis of HCC (Int J Surg. 2018 Jul;55:73-80; Cell Physiol Biochem. 2017;44(3):967-981; Oncotarget. 2016 Jul 19;7(29):45283-45301) .
However, although the molecular mechanisms through which the high level of NLR and PLR are associated with poor HCC outcomes remain unknown, the manuscript discussed this issue and proposed some hypotheses.
Major:
• It would be interesting to compare the preoperative NLR,PLR with postoperative NLR,PLR to evaluate the efficacy of surgery.
• Please supplement more follow-up data of patients to show more detailed results and prognosis.

Minor:
• Line 135-146: the authors combined the NLR and PLR to predict the prognosis. Please explain what is the difference between combination vs. single predictor.
• Line 148-159 the manuscript indicates that the NLR,PLR correlates with tumor size, so can we just choose the tumor size as the predictor instead since tumor size is one of the prognostic predictor for HCC, too? It would be better to add more discussion about the current existing prognostic predictors combining their advantages and disadvantages.

---

## Round 0.2 · Minor Revisions

Please correct English Lines 185-188 and 197-199 (lines refer to PDF version)

Reviewer 1 ·

Basic reporting

This is a revised manuscript. The authors performed a good work.

Experimental design

This is a revised manuscript. The authors performed a good work.

Validity of the findings

This is a revised manuscript. The authors performed a good work.

Additional comments

This is a revised manuscript. The authors performed a good work.

·

Basic reporting

no comment

Experimental design

no comment

Validity of the findings

no comment

Additional comments

Please correct some minor grammar mistakes and improve English writing.

---

## Round 0.3 · Minor Revisions

English in lines 184-186 and 197-198 needs editing.
This editing was already specifically requested in the previous recommendation.

---

## Round 0.4 · accepted · Accept

Thank you for your efforts in language editing.